# Integrated Analysis of Cancer Tissue and Vitreous Humor from Retinoblastoma Eyes Reveals Unique Tumor-Specific Metabolic and Cellular Pathways in Advanced and Non-Advanced Tumors

**DOI:** 10.3390/cells11101668

**Published:** 2022-05-18

**Authors:** Vishnu Suresh Babu, Ashwin Mallipatna, Deepak SA, Gagan Dudeja, Ramaraj Kannan, Rohit Shetty, Archana Padmanabhan Nair, Seetharamanjaneyulu Gundimeda, Shyam S. Chaurasia, Navin Kumar Verma, Rajamani Lakshminarayanan, Stephane Heymans, Veluchamy A. Barathi, Nilanjan Guha, Arkasubhra Ghosh

**Affiliations:** 1GROW Research Laboratory, Narayana Nethralaya Foundation, Bangalore 560099, India; vishnusbabu@narayananethralaya.com (V.S.B.); ramaraj@narayananethralaya.com (R.K.); drrohitshetty@yahoo.com (R.S.); archana.nair@narayananethralaya.com (A.P.N.); 2Department of Cardiology, Cardiovascular Research Institute Maastricht (CARIM), Maastricht University, 6229 ER Maastricht, The Netherlands; 3Retinoblastoma Service, Narayana Nethralaya, Bangalore 560099, India; ashwinmc@gmail.com (A.M.); gagan@narayananethralaya.com (G.D.); 4Agilent Technologies India Pvt Ltd., Bangalore 560048, India; deepak_sa@agilent.com (D.S.); gsranjaneyulu2001@gmail.com (S.G.); 5Ophthalmology and Visual Sciences, Medical College of Wisconsin, Milwaukee, WI 53226, USA; schaurasia@mcw.edu; 6Lee Kong Chian School of Medicine, Nanyang Technological University Singapore, Singapore 308232, Singapore; nkverma@ntu.edu.sg; 7Singapore Eye Research Institute, Singapore 169856, Singapore; lakshminarayanan.rajamani@seri.com.sg (R.L.); amutha.b.veluchamy@seri.com.sg (V.A.B.); 8Centre for Molecular and Vascular Biology, Department of Cardiovascular Sciences, KU Leuven, Herestraat 49, Bus 911, 3000 Leuven, Belgium; 9The Ophthalmology and Visual Sciences ACP, Duke-NUS Medical School, Singapore 169857, Singapore; 10Department of Ophthalmology, Yong Loo Lin School of Medicine, National University of Singapore, Singapore 117597, Singapore

**Keywords:** retinoblastoma, transcriptomics, metabolomics, multi-omics, glycolysis, metabolism, cancer

## Abstract

Retinoblastoma (Rb) is a pediatric intraocular malignancy that is proposed to originate from maturing cone cell precursors in the developing retina. The molecular mechanisms underlying the biological and clinical behaviors are important to understand in order to improve the management of advanced-stage tumors. While the genetic causes of Rb are known, an integrated understanding of the gene expression and metabolic processes in tumors of human eyes is deficient. By integrating transcriptomic profiling from tumor tissues and metabolomics from tumorous eye vitreous humor samples (with healthy, age-matched pediatric retinae and vitreous samples as controls), we uncover unique functional associations between genes and metabolites. We found distinct gene expression patterns between clinically advanced and non-advanced Rb. Global metabolomic analysis of the vitreous humor of the same Rb eyes revealed distinctly altered metabolites, indicating how tumor metabolism has diverged from healthy pediatric retina. Several key enzymes that are related to cellular energy production, such as hexokinase 1, were found to be reduced in a manner corresponding to altered metabolites; notably, a reduction in pyruvate levels. Similarly, E2F2 was the most significantly elevated E2F family member in our cohort that is part of the cell cycle regulatory circuit. Ectopic expression of the wild-type *RB1* gene in the Rb-null Y79 and WERI-Rb1 cells rescued hexokinase 1 expression, while E2F2 levels were repressed. In an additional set of Rb tumor samples and pediatric healthy controls, we further validated differences in the expression of HK1 and E2F2. Through an integrated omics analysis of the transcriptomics and metabolomics of Rb, we uncovered a significantly altered tumor-specific metabolic circuit that reduces its dependence on glycolytic pathways and is governed by Rb1 and HK1.

## 1. Introduction

Retinoblastoma (Rb) is the most common pediatric intraocular tumor, characterized by the presence of retinal lesions with vascularization, vitreous seeding, retinal detachment, and calcification [1]. The major clinical goal in treating children with Rb involves preserving life and vision. It is caused by the loss of the functional retinoblastoma protein (pRb), affecting approximately 8000 children annually [2]. Since pRb is a tumor suppressor protein, bi-allelic mutations in *RB1* and the amplification of oncogenes such as *MYCN* relieve cell-cycle checkpoints, thereby driving neoplastic transformation in precursor retinal cells [3]. *RB1* loss causes various alterations to cellular functions through transcriptional and epigenetic mechanisms [4], a greater understanding of which can provide novel treatment strategies. Although significant advances have been made to understand the genetics [5] and pathogenesis of retinoblastoma [6], it is still the most aggressive and fatal pediatric cancer in developing countries [7,8]. Rb is curable with global salvation strategies [9], and is universally fatal if left untreated. Most Rb tumors are diagnosed at advanced stages, which exhibit resistance to conventional therapy and possess a metastatic risk [10]. Therefore, understanding the molecular mechanism of the Rb tumor subtypes, particularly the molecular milieu in advanced tumors to develop targeted therapies, remains the most important prerequisite for better treatment outcomes [11].

Although there have been numerous reports on specific molecular aspects of retinoblastoma [12], concerted efforts to identify molecular pathways and factors that are conserved across the genetic, transcriptomic, and metabolomic profiles in the same tumor tissues are lacking. Such a unified investigation strategy is well suited to reveal better correlations between genetic, gene expression, and metabolic profiles that can be mechanistically linked to disease progression [13]. In this study, we analyzed the transcriptomic and metabolomic profiles of Rb tumors and vitreous humor to identify markers that are associated with the different stages of the cancer. Our goal was to identify, in human tumor samples, specific markers and enriched pathways that are specific to the cancer stage, which can be used to better understand the disease and to tailor treatment modalities more accurately, an unmet clinical need.

## 2. Materials and Methods

### 2.1. Clinical Samples

The study was conducted by the Declaration of Helsinki principles under a protocol approved by the institutional ethics committee of Narayana Nethralaya (EC Ref. no: C/2013/03/02). Informed written consent was received from all parents before inclusion in the study. Histology confirmed Rb tumors (*n* = 9) comprising Group E and Group D of the age range 0.2–4 years, and pediatric controls (*n* = 2) of the age range 0.2–0.3 years, were used for the microarray and metabolomics study. The details of clinical samples, including age, gender, laterality, tumor viability, and clinical and histopathology details, are mentioned in Table 1. For the immunohistochemistry validations, we have used additional Rb subjects (*n* = 25) comprising Group E and Group D of the age range 0.2–4 years. Clinical and histopathology details are mentioned in Appendix A.

### 2.2. Tumor mRNA Profiling

Total RNA was isolated from 9 Rb tumors and 2 control pediatric retinal samples, using the Agilent Absolutely RNA miRNA kit (cat#400814, Agilent Technologies, Santa Clara, CA, USA) according to the manufacturer’s instructions. For the cell culture microarray, total RNA was isolated from *RB1*-null Y79 and *RB1*-overexpressed Y79 cells, using the Agilent Absolutely RNA miRNA kit. Twenty-five nanograms of RNA was taken from each Rb tumor and each control pediatric retina, and the cells were labeled with Cy3 dye using an Agilent Low Input Quick Amp Labeling Kit (p/n 5190-2305, Agilent Technologies, Santa Clara, CA, USA). Gene expression microarray analysis was performed using the Agilent SurePrint G3 Human GE 8 × 60 K V2 Microarray (Agilent Technologies, Santa Clara, CA, USA) and an Agilent SureScan Microarray scanner. The gene expression data were extracted using Agilent Feature Extraction Software (11.5.1.1) and analyzed using Agilent GeneSpring GX 13.1. The analysis was carried out using a t-test unpaired statistical method with the Benjamini–Hochberg FDR method. 

### 2.3. Metabolomics

The samples from 9 patients and 2 controls were extracted using methanol:ethanol (1:1 *v*/*v*). The extracted samples were subjected to LC/QTOF-MS analysis, using an Agilent 1290 Infinity LC System coupled to an Agilent 6550 Accurate mass QTOF LC-MS system (Agilent Technologies, Santa Clara, CA, USA). The data was acquired using electrospray ionization in positive and negative ion modes using a modified polar reverse-phase C18 column and an HILIC column. Molecular features were detected using Agilent MassHunter Profinder (v. B.06.00), and were searched and confirmed by matching them against the Agilent METLIN MS/MS library. Agilent Mass Profiler Professional software (MPP) was used for the statistical comparison of the LC/MS data from the Rb and control samples.

### 2.4. Multi-Omic Data Analysis

The metabolomics and gene microarray results were combined and analyzed using a pathway-centric approach. Transcriptomics and metabolomics data were co-visualized in a pathway context using the Multi-Omics Analysis tool of GeneSpring GX 13.1. This enabled the identification of the differential pathways and entities across both gene expression and metabolomics. Information related to Agilent Technologies products described in this manuscript is for research use only, and not for use in diagnostic procedures.

### 2.5. Pathway Enrichment Analysis and Regulatory Network Analysis

The KEGG functional enrichment of the tumor microarray, metabolomics, and the cell culture microarray was carried out using the GeneSpring GX 13.1, NetworkAnalyst 3.0, and MetaboAnalyst 5.0 packages; and the pathways with *p* < 0.05 and fold-change > 2.0 were significantly enriched. KEGG pathway enrichment analysis using GeneSpring GX 13.1 was also conducted on the up- and downregulated genes and metabolites that were involved in relationship pairs. Based on the interaction information, the construction of gene interactions was performed using NetworkAnalyst and Cytoscape 3.2. The entire metabolome mapping and gene-metabolite interactions network was constructed using MetaboAnalyst 5.0.

### 2.6. Gene Expression Analysis

Total RNA that was extracted for microarray from the clinical subjects was also used for RT-PCR validation. RT-PCR was performed with Agilent Brilliant III Ultra-Fast RT-PCR reagent (cat#600884, Agilent Technologies, Santa Clara, CA, USA), using Agilent AriaMX real-time PCR instruments. The relative mRNA expression levels were quantified using the C(t) method. For in vitro assays, total RNA was isolated from cells using the Trizol reagent (Invitrogen, Waltham, MA, USA) according to the manufacturer’s protocol. A total of 1 µg of RNA was reverse transcribed using the Bio-Rad iScript cDNA synthesis kit (cat#1708890, Bio-Rad, Hercules, CA, USA), and quantitative real-time PCR was performed using the Kappa Sybr Fast qPCR kit (cat#KK4601, Kapa Biosystems Pty (Ltd.), Cape Town, South Africa) using the Bio-Rad CFX96 system. Relative mRNA expression levels were quantified using the C(t) method. The results were normalized to housekeeping human β-actin. The details of the primers used are described in Appendix A.

### 2.7. Cell Lines

WERI-Rb1 and Y79 cells were obtained from the American Type Culture Collection (ATCC, Manassas, VA, USA). Cells were cultured in RPMI 1640 medium (Gibco, Grand Island, NY, USA, cat#11875093) supplemented with 10% FBS and 1% Pen Strep (Penicillin–Streptomycin), and maintained at 37 °C in a humidified atmosphere of 5% CO_2_, with intermittent shaking in an upright T25 flask.

### 2.8. Histopathology & Light Microscopy

Paraffin-embedded specimens of Rb tumor (*n* = 9) and control retina (*n* = 2) were used. A total of 4 µm paraffin sections were dewaxed at 60 °C and rehydrated in decreasing concentrations of ethanol. Slides were stained with hematoxylin and eosin according to standard procedures. Brightfield images were captured using an Olympus CKX53 microscope and Carl Zeiss Axioplan 2 microscope (Carl Zeiss, Oberkochen, Germany).

### 2.9. Immunohistochemistry

For IHC, 4 µm sections of Rb tumor (*n* = 25) and pediatric retina (*n* = 2) were deparaffinized and rehydrated, and were subjected to heat-induced epitope retrieval using citrate buffer for 20 min at 100 °C. After an endogenous peroxidase block, tissues were incubated overnight at 4 °C with antibodies for Ki67 (1:1000; cat#ab16667, Abcam, Cambridge, UK), Rb (1:500; cat#9309, Cell Signaling Technology, Danvers, MA, USA), phospho-Rb (1:500; cat#8516, Cell Signaling Technology, Danvers, MA, USA), E2F1(1:1000; cat#sc251, Santa Cruz Biotechnologies, Dallas, TX, USA), E2F2 (1:500, cat#209662, Abcam, Cambridge, UK), and HK1 (1:500, cat#ab55144, Abcam, Cambridge, UK). The signals were developed using the DAB substrate (Dako Envision System, Agilent Technologies, Santa Clara, CA, USA). Images were analyzed and captured at Brightfield using an Olympus CKX53 microscope. Following the assessment of the staining using manual scoring, the scores were tabulated as ranks over a range of 0–3, with 0 indicating less than 10% positive cells and 3 indicating more than 80% positive cells.

### 2.10. Lentiviral Plasmid and Vector

We constructed a lentiviral plasmid expressing the *RB1* gene in the pCL20 backbone. The cloning strategy was designed using ApE software (Version 8.5.2.0). The pCL20c vector was digested with the 5′ SmaI site and the 3′ KpnI site to incorporate the *RB1* gene. Forward and reverse primers with compatible restriction sites were designed to separate the *RB1* gene via PCR amplification from pSG5L HA-RB (Addgene, Cambridge, MA, USA). The details of the cloning primers used are:

*RB1* Forward: 5′-CCTACGACGTGCCCGACTACG-3′.

*RB1* Reverse: 3′-AGTGACCGGTTCATTTCTCTTCCTTGTTTGAG-5′.

Post-ligation, the product was transformed using DH5α and the colonies were screened using restriction mapping and confirmed using sequencing.

### 2.11. Western Blotting

For Western blot analysis, cells were lysed in RIPA buffer (20 mM Tris pH 8.0, 0.1% SDS, 150 mM NaCl, 0.08% sodium deoxycholate, and 1% NP40 supplemented with 1 tablet of protease inhibitor (Complete ultra mini-tablet, Roche, Indianapolis, IN, USA) and phosphatase inhibitor (PhosStop tablet, Roche, Indianapolis, IN, USA)). A total of 20 µg of total protein was loaded per lane, and protein was separated using SDS-PAGE. The separated proteins on the gel were transferred onto a PVDF membrane and were probed for specific antibodies against Rb (cat#9390; Cell signaling, Danvers, MA, USA), phospho-Rb (cat#8516, Cell signaling, Danvers, MA, USA), E2F2 (ab209662; Abcam, Cambridge, UK), HK1(cat#2024; Cell signaling, Danvers, MA, USA), and GAPDH (cat#5174; Cell signaling, Danvers, MA, USA) at 1:1000 dilution in 5% BSA in 1× TBST, overnight at 4 °C. After 4 washes with 1× TBST for 10 min, membranes were incubated with HRP-conjugated anti-mouse (cat#7076; Cell signaling, Danvers, MA, USA) or anti-rabbit antibodies (cat#7074; Cell signaling, Danvers, MA, USA) at 1:2000 dilution for 2 h. Images were visualized using the Image Quant LAS 500 system (GE Healthcare Life Sciences, Piscataway, NJ, USA).

### 2.12. Statistical Analysis

A statistical analysis was performed using GraphPad Prism 8. Data are presented as the mean ± S.D. unless indicated otherwise, and *p* < 0.05 was considered as being statistically significant. For all representative images, results were reproduced at least three times in independent experiments. For all quantitative data, the statistical test used is indicated in the legends. A statistical ‘decision tree’ is shown in Appendix A. Heatmaps of the Z-transformed gene expression level of the mRNA microarray were created using Python 3.7 Seaborne 0.9.0.

## 3. Results

### 3.1. Transcriptomic Profiling of Retinoblastoma Tumors Identifies Distinct Expression Profiles in Rb Subtypes

To unravel the molecular networks in Rb tumors, we performed total mRNA profiling using gene expression microarrays in enucleated tumor tissues from nine retinoblastoma patients and two age-matched pediatric retinae as the control (Table 1), as the discovery cohort. We identified distinct dysregulated gene clusters, implicating gross differences between Rb tumors and the controls (Figure 1A). The top upregulated genes identified in the microarray comprised RB1 pathway-related transcription factors [14] such as *E2F1* (*p* < 0.05, FC = 29.2), *E2F2* (*p* < 0.05, FC = 540.2), and key cell cycle checkpoint genes [15] such as *CCNB2* (*p* < 0.05, FC = 319.4), *CCNE2* (*p* < 0.05, FC = 46.81), *CDK1* (*p* < 0.05, FC = 39.81), *CDKN2A* (*p* < 0.05, FC = 29.28), and *CHEK2* (*p* < 0.05, FC = 26.9). We also identified the significant upregulation of immune system-related genes such as *CD86* (*p* < 0.05, FC = 493.5) and *CD19* (*p* < 0.05, FC = 13.27) [16], and epigenetic regulators such as *SYK* [17] (*p* < 0.05, FC = 14.9) and *PRDM1* [18] (*p* < 0.05, FC = 15.89) in Rb tumors. Strikingly, we also identified the mitochondrial TCA-related *FAHD1* gene [19] (*p* < 0.05, FC = 2.38) to be significantly upregulated in Rb subjects. The top downregulated genes comprised photoreceptor-related genes such as *RHO* [20] (*p* < 0.05, FC = −1058.9), *NRL* [21] (*p* < 0.05, FC = −104.00), *PDE6D* [22] (*p* < 0.05, FC = −7.178), *CRABP1* [23] (*p* < 0.05, FC = −140.60), and glycolytic factors such as *HK1* [24] (*p* < 0.05, FC = −17.46), *SLC2A1* [25] (*p* < 0.05, FC = −8.16), and *FOXO3* [26] (*p* < 0.05, FC = −5.08). The methyltransferase *MGMT* [27] (*p* < 0.05, FC = −4.8) and *RB1* (*p* < 0.05, FC = −4.36) were significantly downregulated in Rb tumors. We applied a standard QC procedure to the dataset considering the sample quality, hybridization quality, signal comparability, and array correlation. The boxplot of raw intensities (Appendix A) and the density histogram of the log intensity distribution (Figure 1B) of each array before normalization provide an overview of the dataset quality. Normalization was able to sufficiently remove smaller discrepancies, leading to comparable distributions between all arrays (Figure 1C and Appendix A). The findings prompted us to elucidate the microarray analysis in the Rb subtypes. We segregated the Rb cohort into advanced Rb (defined as AJCC Stage [28] cT3 or IIRC [29]) and non-advanced Rb (defined as AJCC Stage cT2 or IIRC Group D), based on their clinical and histopathological information (advanced Rb, *n* = 5; non-advanced Rb, *n* = 4). Using the clustering analysis functionality of GeneSpring GX 13.1 (Agilent Technologies, Santa Clara, CA, USA), we found that the advanced Rb subjects (P1–P5) strongly correlated with each other, compared to the non-advanced (P6–P9) and the controls (Appendix A). However, advanced Rb subjects exhibited a weak correlation with the controls in the microarray (Figure 1D, r_s_ = 0.42), which is suggestive of their greater de-differentiation state, while non-advanced Rb subjects showed a better correlation with the controls (Figure 1E, rs = 0.9). We identified distinct clusters of gene expression profiles between the advanced Rb, non-advanced Rb, and the controls (Figure 1F). In the advanced Rb tumors, we identified 6089 genes using microarray, of which 1027 gene sets were unique to the clinical pathology-defined high-risk tumor (*p* < 0.05, FC > 2). We identified 2633 genes that were unique to the non-advanced Rb subtype out of 7695 genes detected in the microarray (*p* < 0.05, FC > 2), compared to age-matched healthy retinae (Figure 1G).

### 3.2. Rb Tumor Clinical Subtypes Demonstrate Altered Molecular Pathways Unique to Their Stage

We compared the transcriptomic profiles of two Rb subtypes and identified a set of significant genes belonging to the top enriched pathways in the Rb subtypes (*p* < 0.005). Among the previously identified genes in Rb [2], several were found to be differentially expressed in advanced and non-advanced Rb (*E2F’s*, *CDK’s*, *MYCN*, and *SYK*). We further arranged differentially expressed significant genes in advanced and non-advanced Rb into four gene groups showing their interaction networks (Figure 2A). Gene group 1 genes were significantly upregulated in advanced Rb tumors, compared to non-advanced Rb and controls (*CDK1*, *CDKN2A*, *CCNB2*, *CCNE2*, *PTTG1*, and *DDB1*). We performed KEGG enrichment analysis on gene group 1 using the gene sets from the gene ontology biological processes (GOBP) and the MSigDB hallmarks (HALLMARK), and revealed the gene interaction network of group 1 genes (Figure 2B) and their functional overlapping pathways (*p* < 0.005). Group 1 genes were associated with pathways such as cell cycle regulation, mitotic spindle checkpoint, DNA damage and response, and cytokinesis (Figure 2C). Gene group 2 displayed a significant upregulation of E2F transcription activators (*E2F1, E2F2*, *E2F3*, and *E2F6*) [30] and tumor-specific transcription factors (*MYCN* [31], *RUNX1* [32], and *GABPB1* [33]) in advanced Rb. Strikingly, we found a significant downregulation of transcription factors such as *FOXO3* and repressor E2Fs (*E2F7* and *E2F8*) [30] in the advanced Rb cohort compared to non-advanced tumors and controls. The epigenetic regulators show a high degree of expression for *SYK*, *PRDM1*, and *TK1* in advanced Rb, while relatively lower expression of these factors was detected in non-advanced Rb. Notably, advanced Rb displayed a low degree of expression of *BRD4*, *DNMT3A*, and *MGMT*, while non-advanced Rb displayed a high degree of expression of these epigenetic factors. Immune-related genes such as *CD86*, *CD19*, and *CD36* were significantly upregulated in advanced Rb while displaying a low degree of expression of *CD81* and *CD163*. KEGG enrichment analysis revealed gene interaction networks of group 2 genes (Figure 2D) and their overlapping pathways (*p* < 0.005). Gene group 2 shows an association with pathways including cellular senescence, B-cell receptor signaling, pathways in cancer, transcriptional misregulation, JAK/STAT signaling, E2F targets, and MYC targets (Figure 2E). Gene group 3 shows differential expression of metabolic genes related to glycolysis, Krebs cycle, and fatty acid metabolism. Notably, glycolytic genes displayed a significantly low degree of expression in advanced and non-advanced subjects compared to the controls (*HK1*, *HK2*, *HK3*, *GCKR*, *SLC2A1*, *G6PC*, *ALDOC1*, and *ENO1*). However, Krebs cycle-related genes were significantly altered across all Rb subjects compared to the controls. Advanced Rb subjects showed a significantly high degree of expression for fatty acid metabolism genes such as *MYCLD* and *ACSL1*, compared to non-advanced Rb tumors. Notably, the PPAR pathway-related genes (*PPARGC1A*, *PPARA*, and *PPARD*) displayed a low degree of expression in advanced and non-advanced Rb tumors compared to the controls. KEGG enrichment analysis revealed gene interaction networks of group 2 genes (Figure 2F) and their overlapping pathways (*p* < 0.005). Group 3 genes were associated with pathways such as the integration of energy metabolism, glutamine biosynthesis, the regulation of lipid metabolism, pyrimidine salvage, ETC, and glycolysis (Figure 2G). Gene group 4 shows the enrichment of phototransduction specific genes that were significantly low in advanced and non-advanced Rb tumors. Network analysis shows the interaction of RHO with *CRX* and *NRL*, and the functional pathways regulated by the Group 4 genes (Appendix A). Thus, the analysis revealed several previously unreported gene expression clusters and the associations that are unique to the Rb tumor stage.

### 3.3. Quantitative Validation of Microarray-Identified Targets Confirm Distinct Transcriptomic Profiles in Rb Subtypes

We performed RT-PCR to further evaluate the expression patterns of microarray-identified targets in advanced and non-advanced Rb groups. We found a significant degree of upregulation of cell cycle checkpoint genes such as *CDK1*, *CCNB2*, *CCNE2*, *CDKN2A*, *CDKN3*, *PTTG1*, *CHEK2*, and *RPA4* (*p* ≤ 0.05) in advanced Rb tumors compared to non-advanced Rb tumors (Figure 3A). Microarray-identified epigenetic targets including *PRDM1*, *SYK*, and *TK1* were also found to be upregulated in an RT-PCR analysis of advanced Rb subjects compared to non-advanced tumors (Figure 3B). Key photoreceptor genes such as *RLBP1*, *RDH12*, *CRABP1*, *NRL*, *SOX8*, *RHO*, *PAX6*, and *SAG* were significantly low in advanced Rb tumors, compared to non-advanced Rb tumors in RT-PCR analysis (*p* ≤ 0.05), further confirming the gene expression pattern identified in the tumor microarray (Figure 3C). RT-PCR analysis further confirmed the microarray expression patterns of glycolytic genes such as *HK1* and *G6PC*, which shows significant downregulation in advanced Rb compared to non-advanced Rb (*p* ≤ 0.05) (Figure 3D). RT-PCR analysis confirmed the microarray expression of the E2F transcription activator *E2F2* to be significantly high in advanced Rb tumors, compared to non-advanced Rb tumors. The E2F transcription repressor *E2F7* was significantly low in advanced Rb tumors, compared to non-advanced Rb (*p* ≤ 0.05), further confirming the microarray-identified expression pattern (Figure 3E). RT-PCR log2-normalized values of the validated set of genes associated with the above-mentioned pathways showed a strong correlation with log2-normalized values of the same set of genes identified using the microarray analysis (r = 0.61, *p* < 0.001) (Figure 3F).

### 3.4. Differentially Accumulated Metabolites Reveal the Enrichment of Key Metabolic Pathways in Rb Vitreous Humor

The transcriptomic profile of the Rb tumor prompted us to elucidate the differential metabolites in the vitreous humor samples of the same Rb subjects and controls using metabolomics analysis. The principal component analysis of the samples shows a marked difference between Rb subjects and controls (Figure 4A). Metabolomics analysis was conducted using the LC-MS positive and negative (HILIC and C18 columns) method for a broader coverage of metabolites. A total of 1190 metabolites were detected in LC-MS analysis and the data were normalized to reduce or eliminate the effects of total sample amount variation on the quantification of individual metabolites (Figure 4B and Appendix A). We performed a pairwise analysis of the samples using the clustering analysis functionality of GeneSpring. The correlation analysis followed by clustering showed a strong relationship between advanced Rb subjects (P1–P5) compared to non-advanced Rb (P6–P9) and controls (Appendix A). Statistical analysis of the metabolomics data with stringent filters (*p* < 0.05, FC > 2) revealed 350 differentially expressed metabolites in the Rb vitreous humor (Figure 4C). Hierarchical clustering analysis performed on differentially accumulated metabolites identified components of lipid metabolism (sphingosine and ceramide) and fatty acid metabolism (okadaic acid, methyl palmitate, decanoyl-CoA, phytanic acid, and hexacosanoic acid) to be upregulated in advanced Rb compared to the non-advanced group. Strikingly, we found a low expression of soraphen A, an ACC inhibitor [34], in the advanced group, further highlighting the role of long-chain fatty acid synthesis in high-risk Rb tumors. A subset of the amino acids, namely creatine, taurine, and isoleucine–histidine–lysine displayed varied degrees of regulation in Rb subtypes. However, key metabolites of glycolysis (pyruvic acid, lactobionic acid, and galactosamine) were significantly downregulated in advanced Rb (Figure 4D). KEGG pathway analysis showed the enrichment of metabolites associated with linoleic acid metabolism, taurine, and hypo-taurine metabolism, and tyrosine metabolism, etc., as the top targets in Rb vitreous. Notably, metabolites that were associated with the citrate cycle and fatty acid metabolism showed a higher degree of enrichment than glycolysis and pyruvate metabolism in Rb vitreous (Figure 4E). This analysis highlights the unique details of significantly altered tumor metabolism in Rb eyes compared to healthy ones.

### 3.5. Integrated Transcriptomic and Metabolome Analysis Reveals Gene–Metabolite Interaction Networks Associated with Retinoblastoma

To gain insights into the overlapping pathways in Rb, we performed an integrated network analysis of transcriptomics and metabolomics to identify unique functional relationships between gene expression and metabolism. Using the KEGG global metabolic network, we mapped differentially accumulated metabolites detected in Rb vitreous samples (Figure 5A). Further, the enrichment analysis module of MetaboAnalyst was used, which further confirmed the significant alteration of fatty acid metabolism (yellow edges), amino acid metabolism (violet edges), and the pentose phosphate pathway (green edges) in the global metabolome map of Rb (Figure 5A). The metabolome map also highlighted the relatively low activity of glycolysis (blue edges) and the low abundance of the pyruvate metabolite. We constructed a metabolite–gene interaction network by integrating the differentially expressed transcriptomic targets and the associated metabolites (Figure 5B). The interactive map highlights the low abundance of the pyruvic acid metabolite (*p* < 0.05, FC < 2) and the significant downregulation of glycolytic genes (*p* < 0.05, FC < 2). Similarly, a low abundance of valproic acid (*p* < 0.05, FC < 2) is linked to a significant upregulation of *HDAC2* (Histone Deacetylase 2) gene expression (*p* < 0.05, FC > 2) in the interactome map. The *HDAC2* gene plays a redundant role in lipid accumulation [35], further confirming the high fatty acid metabolic state in Rb. A relatively high expression of *FN1* (fibronectin1) and *BAAT* (Bile Acid-CoA:Amino Acid N-Acyltransferase) was also highlighted with the low abundance of the taurine metabolite in the interactome map (Figure 5B). *FN1* overexpression can trigger ER stress and facilitate lipid accumulation [36], while *BAAT* transcription in conjugation with glycine or taurine regulates cholesterol and phospholipid synthesis [37], favoring cancer growth. Highly abundant metabolites such as salicylic acid, creatine, and myristic acid (*p* < 0.05, FC > 2) were found to interact with downregulated metabolism-associated genes such as *HIF1A* (Hypoxia inducible factor 1 subunit alpha), *PRKAA2* (Protein Kinase AMP-Activated Catalytic Subunit Alpha 2), and *GUCA1A* (Guanylate Cyclase Activator 1A) (*p* < 0.05, FC < 2). A low expression of *HIF1A* reduces glycolytic metabolism and enhances mitochondrial oxidative phosphorylation [38,39], further providing evidence for low glycolysis in Rb. *PRKAA2* encodes the AMPKα protein, which functions as a metabolic sensor in many diseases, including cancer [40]. The *GUCA1A* gene encodes for the GCAP1 protein, which regulates N-terminal myristoylation and lipid modifications in photoreceptors [41]. A combined pathway analysis of transcriptomics and metabolomics identified glycolysis, AMPKα signaling, HIF-1 signaling, and fatty acid biosynthesis to be altered in Rb tumors (Appendix A).

### 3.6. Validation of Molecular Signatures That Specify Clinical and Histopathological Grades of Retinoblastoma

To validate the multi-omics-identified targets in Rb, we first performed immunohistochemical staining on the same set of advanced and non-advanced Rb tumor tissues, using pediatric retina as the controls to evaluate the classical molecular signs. H and E staining confirmed densely packed tumor cells with little cytoplasm partly arranged in perivascular cuffs in advanced Rb (Figure 6A). In contrast, the non-advanced Rb tissues contained tumor cells surrounding cystic spaces with relatively more cytoplasm and smaller nuclei. Healthy pediatric retina shows clear and visible retinal layers in H and E staining. Advanced Rb subjects showed strong and consistent immunostaining positivity for the proliferation marker Ki67 and the transcription factor E2F1, while their signals were relatively weaker, though not significant in non-advanced Rb, and undetectable in the pediatric retina (Figure 6A,C). The total Rb expression, observed in the healthy retina, was not detectable in tumors using IHC (Figure 6B,C). We further profiled the microarray expression of *Ki67*, *E2F1*, *RB1*, *RBL1*, and *RBL2* in Rb subjects based on their laterality and severity. *Ki67* expression was higher in bilateral and unilateral Rb tumors, and this consistent expression trend was evident in advanced and non-advanced Rb subjects (Figure 6D). *E2F1* expression was higher in the bilateral group compared to the unilateral (*p* < 0.005), and in advanced Rb compared to non-advanced Rb (*p* < 0.05), providing clues as to Rb-loss mediated transcriptional effects. *RB1* expression was significantly low in unilateral Rb compared to bilateral, advanced, and non-advanced Rb groups (Figure 6F), while its pocket protein family gene *RBL2* (p130) [42] was significantly low in the bilateral and unilateral Rb groups, and in the advanced and non-advanced Rb compared to the pediatric controls (Figure 6H). However, *RBL1* (p107) was upregulated in Rb subjects, confirming its cell cycle compensation effects [43] compared to the controls (Figure 6G).

### 3.7. Validation of Multi-Omics Findings in Rb Subjects and the In Vitro Model

The combined analysis of transcriptomics and metabolomics revealed glycolysis and pathways in cancer to be significantly altered in Rb (*p* < 0.001). We corroborated these findings by validating the expression of their key candidate markers: HK1, which regulates glycolysis [24], and the E2F2 transcription factor associated with key cancer pathways [44]. We performed an immunohistochemical analysis on an additional cohort of advanced (*n* = 15) and non-advanced Rb tissues (*n* = 10), and detected strong and consistent immunostaining positivity for E2F2 (IHC score= 2–3), while the signals were undetectable in control pediatric retinae (*n* = 2, IHC score = 0). HK1 showed weak signals in advanced (*n* = 15) and non-advanced Rb (*n* = 10) tumor areas (IHC score = 0–1), while strongly staining the intact retina portions within non-advanced Rb tissues, thereby serving as an internal control. Notably, HK1 exhibited strong and consistent signals in the photoreceptor layer of healthy pediatric controls (*n* = 2; IHC score= 3) (Figure 7A,B). RT-PCR validations further confirmed the expression pattern of *E2F2* and *HK1* in advanced and non-advanced Rb tumors (Figure 7C). To elucidate how *RB1* influences *HK1* and *E2F2* expression in the context of cancer, we overexpressed the wild-type *RB1* in Rb null WERI-Rb1 retinoblastoma cells in vitro. The ectopic expression of RB1 induced HK1 protein and transcript in WERI-Rb1 cells (Figure 7D,E,G, *p* < 0.001), while reducing E2F2 transcript and protein levels (Figure 7D, F, *p* < 0.001). We further performed total mRNA profiling using *RB1*-null Y79 and *RB1*-overexpressed Y79 microarrays. We applied the standard QC procedure, and the data was normalized, leading to comparable distributions between all arrays (Appendix A). Statistical analysis revealed a distinct cluster of expression profiles between *RB1*-null and *RB1*-overexpressed Y79 (Appendix A). We identified 65 differentially expressed genes between *RB1*-null and *RB1*-overexpressed Y79 (*p* < 0.05, FC > 2). The E2F transcription factor genes (*E2F1*, *E2F2*) and cell cycle genes (*CDK1*, *CCNB2*) were significantly upregulated (*p* < 0.05, FC > 2), and the glycolytic genes were significantly downregulated (*HK1*, *ENO1*, and *G6PC3*) (*p* < 0.05, FC < 2) (Figure 7H). The transcriptomic analysis also revealed a high degree of expression for the aldolase isoenzymes *ALDOA*, *ALDOB*, and *ALDOC* in the Rb-null condition, while *RB1* complementation showed a significant downregulation of aldolase isoenzymes (Figure 7I). KEGG enrichment pathway analysis revealed sphingolipid metabolism, branched-chain amino-acid metabolism, apoptosis, and the DNA repair pathway to be significantly altered between the *RB1* null and *RB1*-complemented Y79 cells, indicating that the cell line microarray had significant similarities with the transcriptomic landscape of the Rb subjects (Figure 2).

## 4. Discussion

This integrated omics analysis of the Rb tumor transcriptome and the vitreous humor metabolome within the human Rb tumor subtypes using the pediatric retina as controls was performed to enrich our understanding of altered gene function with changes in tumor metabolism. Although there have recently been a few reports on alterations in the Rb tumor-associated metabolites [45,46], we found significantly more functional associations between the metabolic targets and the enriched pathways owing to the coordinated samples used for analysis and the use of age-matched pediatric controls. Although we have focused on intraocular advanced and non-advanced retinoblastoma tumors, our findings can be extended to other cancer systems with persistent *RB1* mutations.

Rb is a complex disease with predominant genetic and epigenetic events [47], and it is important to focus on improving our understanding of the regulatory mechanisms that promote tumor growth and intra-tumoral heterogeneity. To accomplish this, we classified our Rb cohort based on the AJCC or IIRC guidelines as advanced and non-advanced Rb, and used age-matched pediatric controls. The different clinical and pathological features of the two subtypes identified the relevance of this classification. The limitation of our study was the relatively small sample size due to the prerequisite of different sample types that were collected at the same time from each eye in the investigation. However, the major strength of the study, in comparison with the previous publications, is the ability to integrate different omics data sets from tumors and vitreous humors from the same patients. Further, we validated the protein levels in FFPE sections of the same tumors, as well as an additional independent cohort of 25 Rb tissues. These data sets allowed us to discover new insights into the interactions between the data sets, and to reveal new insights into the disease mechanism that was lacking in previously reported single omics studies [48,49,50].

Using transcriptomic profiling, we found there was a high degree of expression of cell cycle regulators and DNA damage and response checkpoint genes in both Rb subtypes, a consequence of pRb loss in the tumors [51,52]. Moreover, we report on the significant enrichment of E2F pathway genes in advanced Rb compared to non-advanced Rb, highlighting major flaws in transcriptional events due to RB1 inactivation in high-risk tumors. In contrast to previous findings on *E2F1* [53], we report on the significantly high expression of *E2F2* in advanced Rb tumors, along with the low expression of the E2F repressors *E2F7* and *E2F8*, compared to non-advanced Rb in our study, which is also supported by a recent study [54]. Various studies have highlighted *MYCN* expression in Rb tumors [55], while we show the presence of the differential expression of *MYCN* in Rb tumor subtypes. Furthermore, we report on the elevation of tumor-specific transcription factors such as *RUNX1* and *GABPB1*, and the reduced expression of *FOXO3* that is low in the advanced group. While these transcriptional alterations are expected from the literature on other high-risk pediatric tumors [56,57,58,59], their distinct transcriptional regulation in advanced Rb tumors is a new observation.

Previous studies have highlighted the role of epigenetic modifications and *SYK* in targeting Rb tumorigenesis [4]. In agreement with these findings, we report on the high expression of *SYK* in advanced Rb tumors compared to non-advanced Rb. In addition, we also discovered the differential expression of *PRDM1* and *TK1* in Rb subtypes, which are known epigenetic regulators in other cancers [60,61]. Our study also highlights the low expression of known canonical epigenetic readers such as *BRD4*, *DNMT3A*, and *MGMT* in advanced Rb tumors, highlighting a diverse epigenetic landscape in advanced tumors, extending our insights into Rb pathogenesis. These observations give support for the further investigation of broad epigenetic modulators for adjunctive treatments in advanced Rb. We further report on the differential expression of several immune cell surface receptors within the tumor tissues. B-cell maturation genes such as *CD19* and its binding partner *CD81* were significantly altered in the Rb subtypes, which have roles in regulating the immune response and receptor signaling [62]. We also observed high *CD86* levels in the advanced tumor, which are supported by the presence of high *CD86* B cell signatures in advanced stages of other cancers [63]. However, a high degree of *CD80* expression, along with *CD86* and increased transcript numbers of T-cell counter receptors (*CD28* and *CTLA4*) (data not shown) in tumors also point out the costimulatory roles of CD86/CD80, which are known to induce T cell anergy [64]. The differential expression of these immune markers highlights a unique immunophenotype of the Rb tumors that requires further exploration to understand their roles in tumor growth and its progression of severity. We also observed a differential loss of visual cycle-associated genes among the clinical subgroups and compared to controls, aspects of which are supported by certain studies [65,66]. Both the Rb subtypes expressed the cone-specific markers *RXR γ*, *CRX*, *OTX1*, and *TRβ2* (data not shown), compared to the pediatric retina, supporting the cone photoreceptor origin of cancer [67].

We found that advanced Rb was associated with a low expression of glycolytic genes and altered Krebs cycle with a higher propensity for fatty acid metabolism. Our findings are also supported by a recent study that highlighted the role of lipid metabolism in Rb tumors [45]. Furthermore, our global metabolomic profiling of Rb vitreous humor identified a variety of differentially accumulated metabolites that are distinct from the pediatric controls. Importantly, we observed a significantly low abundance of pyruvate in Rb vitreous humor, which directly correlated with the loss of glycolytic gene expression, particularly *HK1*. Pathway analysis further confirmed our findings, highlighting the low enrichment of glycolysis and pyruvate metabolism pathways compared to high fatty acid and lipid metabolism in advanced Rb. Therefore, the metabolomics findings in vitreous humor corroborated with the transcriptomics profile of the Rb tumor and revealed a significant distinction in metabolite abundance between the Rb subtypes. The data integration of transcriptomic and metabolomic profiles of Rb revealed altered pathways, including cellular metabolism and transcriptional machinery, and we identified *HK1* and *E2F2* as being representative genes that regulate distinct transcriptional networks in Rb tumors. Through immunohistochemistry, we show the distribution of HK1 and E2F2 proteins in Rb tissue subtypes. The *RB1* complex binds to *E2F2* promoters and represses its function during the development and differentiation programs [68], and we assume that the high expression of *E2F2* in tumors is a direct consequence of *RB1* inactivation. However, *HK1*, which controls glycolytic flux [69], has been found to be upregulated in other tumors [70] and is associated with oncogenes such as *KRAS* [71] mutants. In this aspect, we find Rb tumors to behave in a metabolically opposite manner to other types of solid tumors, thus providing a potentially unique therapeutic target. In addition, recent reports highlight the glycolytic control exerted by pRb proteins [72], and we hypothesize that the low enrichment of glycolysis in tumors is a direct consequence of the loss of pRb proteins. Therefore, our observation in Rb tumors is critical in terms of delineating the unique metabolic phenotypes of tumors of diverse origins. Using in vitro models, we show that *RB1* complementation enhanced the glycolytic HK1 protein and its transcript, while reducing E2F2 levels, further confirming our findings in human tumors. The transcriptomic profiling of RB1-complemented Y79 shows differentially regulated cell cycle and metabolic pathways that corroborated with findings in Rb tumors.

## 5. Conclusions

In conclusion, the comprehensive transcriptomic profiling of the Rb tumor subtypes revealed differentially expressed genes belonging to cell cycle, metabolism, epigenetics, immune regulation, and phototransduction pathways. The global metabolomic analysis of Rb vitreous humor revealed metabolites that are associated with critical signaling pathways that are committed to the cancer state, which are otherwise not apparent. The integration of multi-omics data identified co-expression network modules within Rb clinical subtypes that may be crucial for prognostication and future drug development efforts.

## Figures and Tables

**Figure 1 cells-11-01668-f001:**
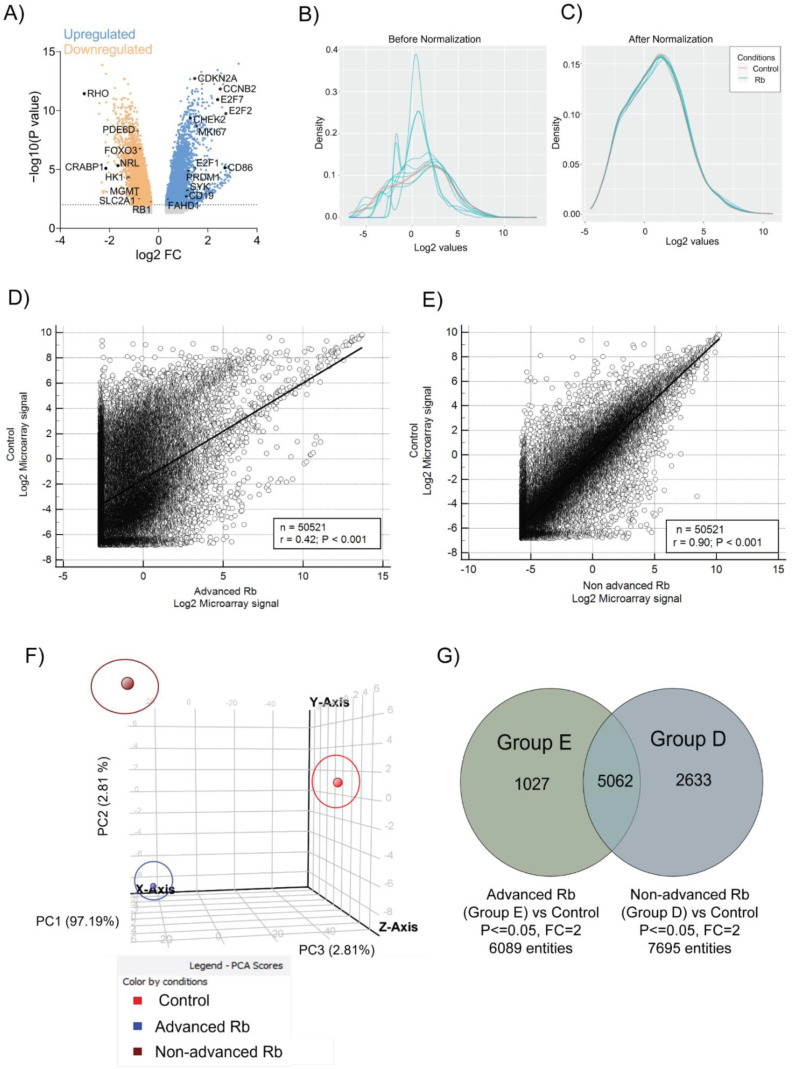
Identification of distinct transcriptional signatures in advanced and non-advanced Rb tumors from primary enucleated eyes. (**A**) Volcano plot of differentially expressed genes. Density curve plot showing the overall signal distribution of all probe sets on the microarray. (**B**) Before normalization. (**C**) After normalization. Spearman’s correlation analysis of genes detected in microarray platform across: (**D**) Control vs. Advanced Rb (r = 0.42, *p* < 0.001); (**E**) Control vs. Non-advanced Rb (r = 0.90, *p* < 0.001. (**F**) Principal component analysis on the microarray results shows distinct clusters, implicating gross differences between the advanced Rb, non-advanced Rb, and the controls. (**G**) Venn diagram showing differential genes identified in Group E (advanced) and Group D (non-advanced) Rb tumor microarrays compared to pediatric retina.

**Figure 2 cells-11-01668-f002:**
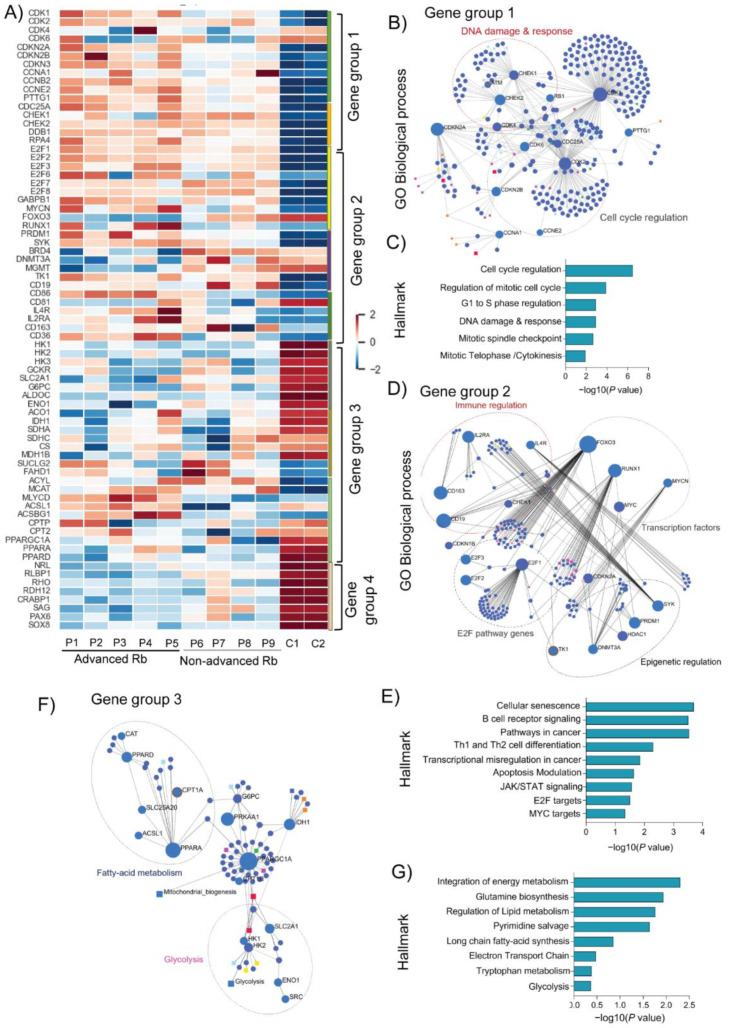
Distinct genes and biological processes are altered in advanced and non-advanced retinoblastoma. (**A**) Heatmap showing significantly differentially expressed genes identified four main gene groups in advanced (*n* = 5) and non-advanced Rb (*n* = 4), compared to pediatric retina controls (*n* = 2). Genes were grouped based on the pathways reported in GOBP. Results are represented as a network of enriched genes sets (nodes) connected to their overlapping pathways (edges). The node size is proportional to the total number of genes in the gene set of interest. (**B**) Gene group 1 represents an enriched cluster of genes in the cell cycle, and the DNA damage and response pathway. (**C**) List of functional pathways regulated by the gene sets in Group 1. (**D**) Gene group 2 represents an enriched group of genes belonging to transcriptional regulation, immune system regulation, and epigenetic factors. (**E**) List of functional pathways regulated by the gene sets in group 2. (**F**) Gene group 3 represents an enriched set of genes belonging to cellular metabolic processes such as glycolysis, Krebs cycle, and fatty acid metabolism. (**G**) List of functional pathways regulated by the gene sets in group 3.

**Figure 3 cells-11-01668-f003:**
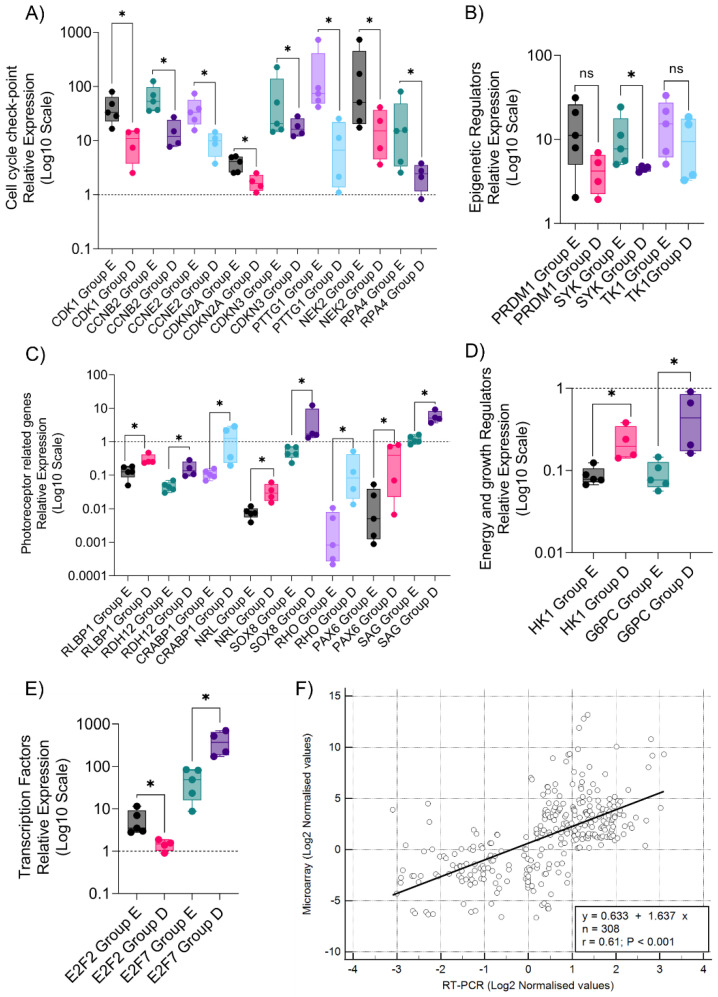
Validation of selected genes for mRNA expression in retinoblastoma subtypes. RT-PCR validation of microarray-identified targets (**A**) Cell cycle checkpoint genes. (**B**) Epigenetic regulators. (**C**) Photoreceptor-specific genes. (**D**) Energy and growth regulator genes. (**E**) Transcription factor genes. (**F**) Pearson correlation plot between microarray and RT-PCR expression of genes involved in cell cycle checkpoint, epigenetic regulation, photoreceptor-specific, energy regulation, and transcription factors. Values represent mean ± SEM. Unpaired two-sided Student’s *t*-test was used for statistical analysis. * *p* < 0.05, ns = no significance.

**Figure 4 cells-11-01668-f004:**
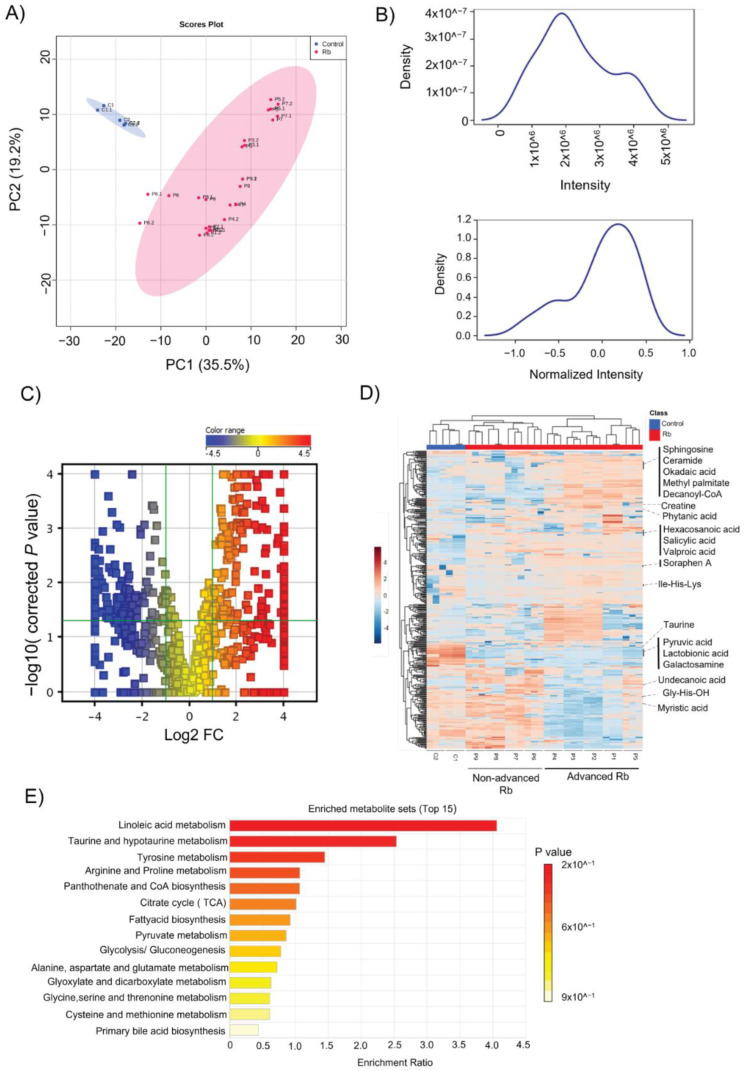
Differentially accumulated metabolites revealed the enrichment of key metabolic pathways in Rb vitreous humor. (**A**) Principal component analysis of metabolite profiling in the vitreous humor of Rb and pediatric controls. (**B**) Density plot showing the distribution of metabolite intensity before and after normalization. (**C**) Volcano plot of differentially accumulated metabolites. *p* < 0.05 (in red), FC > 2: significantly increased; *p* < 0.05, FC < 2 (in blue): significantly reduced relative to the control. (**D**) Hierarchical clustering heatmap of significant metabolites in the vitreous humor of Rb and controls using Pearson correlation as the metric distance. The color code indicates the metabolite’s abundance. (**E**) KEGG pathway enrichment analysis of differentially accumulated metabolites.

**Figure 5 cells-11-01668-f005:**
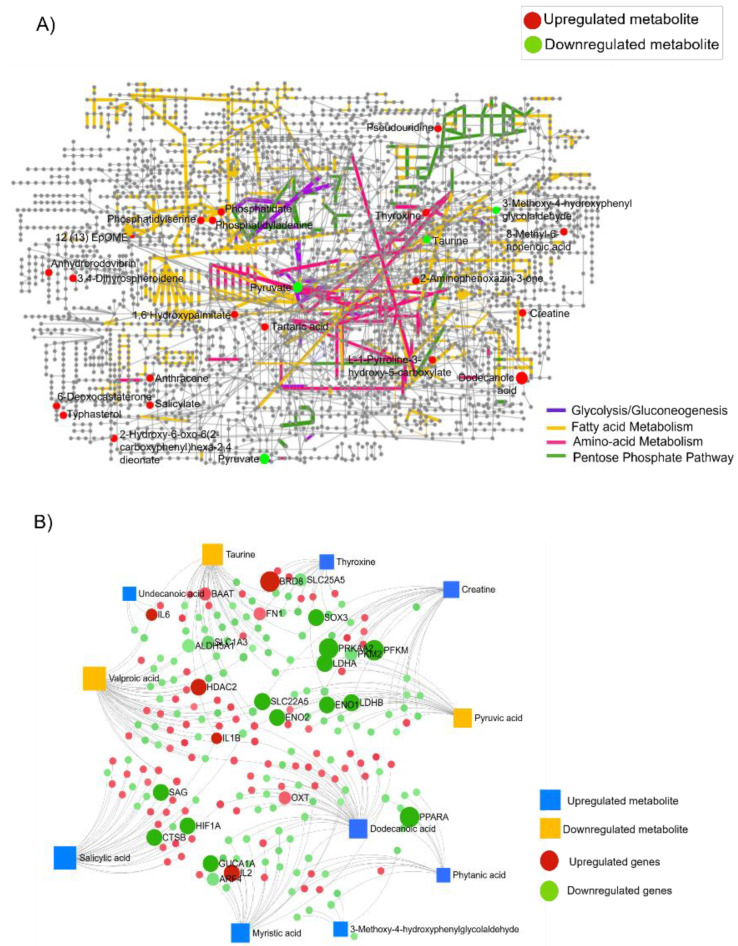
Integrated transcriptomic and metabolome analysis provides unique insights into pathways that are associated with retinoblastoma. (**A**) Differentially accumulated metabolites identified in Rb vitreous were mapped onto the KEGG global metabolome map associated with Rb. The nodes in the figure represent metabolic compounds. Edges are enzymatic transformations. Highlighted nodes represent the differentially accumulated metabolites identified in Rb vitreous. *p* < 0.05, FC > 2; upregulated (in red), *p* < 0.05, FC < 2; downregulated (in green). Edges highlighted in violet represent the glycolysis/gluconeogenesis map in Rb. Edges highlighted in yellow represent the fatty acid metabolism-related enzyme map in Rb. Edges in pink represent the amino-acid metabolism map in Rb. Edges in green represent the pentose phosphate pathway enzymes in Rb. (**B**) Integration of transcriptomic and metabolomic interaction network. Circular nodes indicate differentially regulated genes identified from the microarray; (in green as downregulated set: *p* < 0.05, FC < 2; in red as upregulated set: *p* < 0.05, FC > 2. Square blue nodes indicate differentially accumulated metabolites from metabolomics analysis (*p* < 0.05). Blue square nodes indicate upregulated metabolites (FC > 2), and yellow square nodes indicate downregulated metabolites (FC < 2).

**Figure 6 cells-11-01668-f006:**
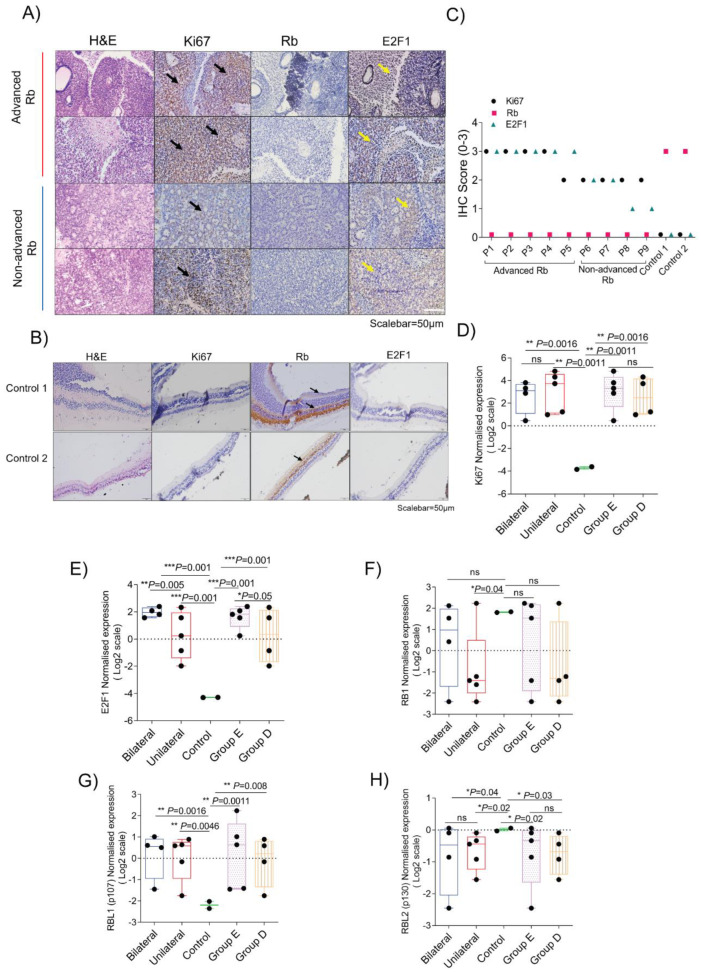
Molecular signatures specific to the clinical and histopathological grades of retinoblastoma in primary enucleated eyes. (**A**) H and E and immunohistochemistry profile showing the expression of Ki67, Rb, and E2F1 on 4 Rb tissues. High levels of expression for Ki67 and E2F1 were observed in the advanced Rb group, while expression of both proteins were lower in the non-advanced Rb group. Scalebar = 50 µm. (**B**) H and E and immunohistochemistry profile showing expression of Ki67, E2F1 on 2 pediatric retina tissues. Rb expression was high in pediatric control tissues while no E2F1 expression was observed. Scalebar = 50 µm. (**C**) IHC scores of Ki67, Rb, and E2F1 in 9 Rb tissues and 2 pediatric retina tissues. (**D**) Gene expression of Ki67 in bilateral, unilateral, Group E, and Group D identified in microarray compared to control. Ki67 expression was high in the Rb tumor microarray. (**E**) Gene expression of *E2F1* in bilateral, unilateral, Group E, and Group D identified in microarray compared to control. *E2F1* expression was significantly higher in Bilateral and Group E Rb subjects compared to unilateral and Group D Rb subjects. (**F**) Gene expression of RB1 in bilateral, unilateral, Group E, and Group D identified in microarray compared to control. *RB1* expression was found to be significantly low in both Group E and Group D compared to pediatric controls. *RB1* expression was low in the unilateral and bilateral groups compared to pediatric control. (**G**) Microarray expression of *RBL1* (p107) in Group E and Group D, unilateral and bilateral Rb subjects. *RBL1* expression was found to be significantly high in both Group E and Group D compared to pediatric controls. *RBL1* expression was high in the unilateral and bilateral group compared to pediatric control. (**H**) Microarray expression of *RBL2* (p130) in Group E, Group D, unilateral, and bilateral Rb subjects. *RBL2* expression was found to be significantly low in both Group E and Group D compared to pediatric controls. *RBL2* expression was low in the unilateral and bilateral groups compared to pediatric control. Values represent mean ± SEM. Unpaired two-sided Student’s *t*-test was used for statistical analysis. * *p* < 0.05, ** *p* < 0.01, *** *p* < 0.001, ns = no significance.

**Figure 7 cells-11-01668-f007:**
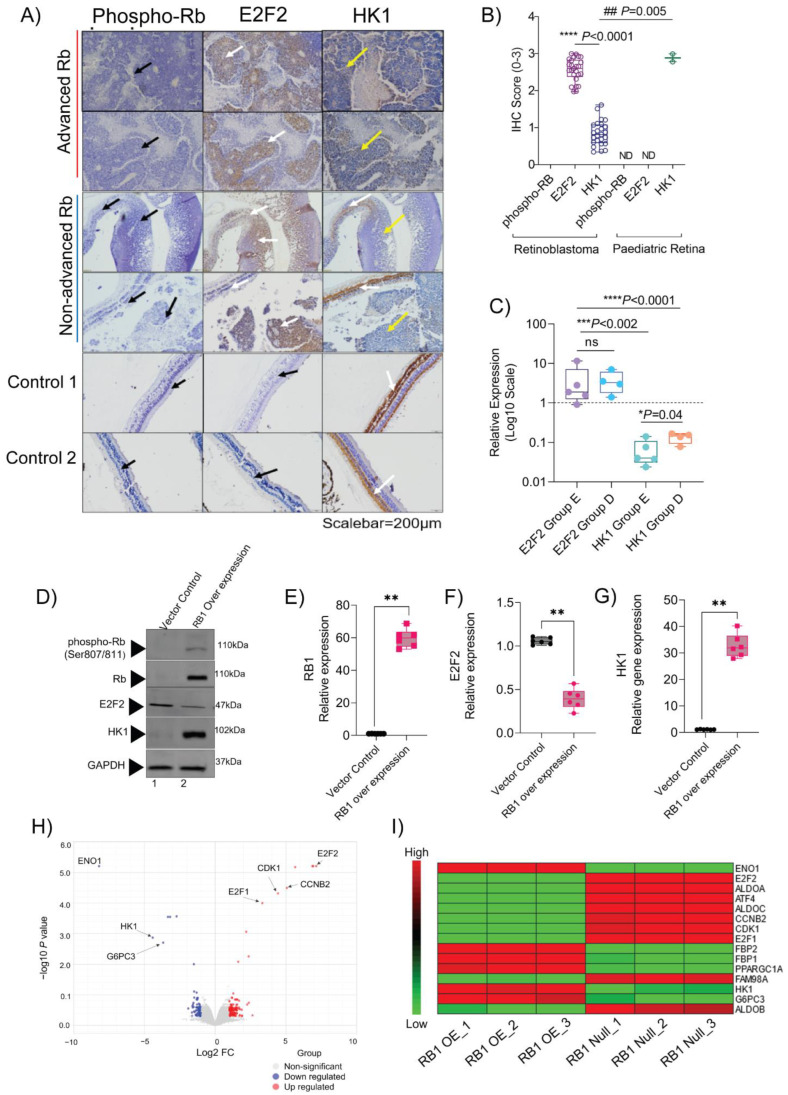
Differentially expressed targets confirm multi-omics findings in enucleated Rb tissue samples and in vitro. (**A**) Immunohistochemistry profile showing the expression of phospho-Rb, E2F2, and HK1 on 4 Rb tissues and 2 pediatric retinae. Scalebar = 200 µm. (**B**) IHC scores of phospho-Rb, E2F2, and HK1 in 25 Rb tissues and 2 pediatric retina tissues. (**C**) Gene expression of *E2F2* and *HK1* in Rb subtypes relative to the pediatric retina. (**D**) Protein expression of phospho-Rb, Rb, E2F2, and HK1 in the control and *RB1*-complemented WERI-Rb1 cells. Relative gene expression of (**E**) *RB1*, (**F**) *E2F2*, and (**G**) *HK1* in control and *RB1*-complemented WERI-Rb1 cells, which illustrate their differences at the molecular level. (**H**) Volcano plot of differentially regulated genes in *RB1*-null and *RB1*-complemented Y79 cells. (**I**) Heatmap showing significantly differentially expressed genes identified in the *RB1*-null and *RB1*-complemented Y79 cells. Values represent mean ± SEM. Unpaired two-sided Student’s *t*-test was used for statistical analysis. * *p* < 0.05, ** *p* < 0.01, *** *p* < 0.001, **** *p* < 0.0001, ## *p* = 0.005.

**Table 1 cells-11-01668-t001:** Clinical and histopathological details of samples.

ID	Sex	Laterality	Age at Presentation	Clinical Risk	IIRC Group	AJCC Staging
P1	M	Bilateral	15 months	Advanced	Group E	cT3b
P2	F	Unilateral	20 months	Advanced	Group E	cT3b
P3	M	Unilateral	24 months	Advanced	Group E	cT3a
P4	F	Bilateral	4 months	Advanced	Group E	cT3b
P5	M	Bilateral	30 months	Advanced	Group E	cT3b
P6	F	Bilateral	21 months	Non-advanced	Group D	cT2b
P7	F	Unilateral	28 months	Non-advanced	Group D	cT2b
P8	M	Unilateral	20 months	Non-advanced	Group D	cT2b
P9	M	Unilateral	21 months	Non-advanced	Group D	cT2a
Control 1	F	NA	3 months	Cardiac Arrest (no ocular complications)		
Control 2	F	NA	2 months	Multiple organ dysfunction (no ocular complications)		

## Data Availability

Data will be available on request.

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
