# Peer review of "Integrated Analysis of Cancer Tissue and Vitreous Humor from Retinoblastoma Eyes Reveals Unique Tumor-Specific Metabolic and Cellular Pathways in Advanced and Non-Advanced Tumors"

_cells, 2022, doi:10.3390/cells11101668_

Round 1

Reviewer 1 Report

The manuscript presents results of transcripts and metabolic pathways analyses in retinoblastoma tumors compared with the normal retina. The assays were performed in tumor biopsies, for RNA transcripts characterization, and in the vitreous humor, for metabolites analysis, obtained from retinoblastoma patients. The microarray method identifies differential expression of genes, encoding for several biological pathways, between clinically advanced and non-advanced tumors. These results were validated by quantitated RT-PCRs. The metabolomic analysis identifies some enriched metabolic pathways in the surrounding tumor media. Metabolic pathways of the humor were compared with the transcripts presented in tumor for the correlation of both and thus, for obtaining an integrated analysis. The analysis of gene expression was also performed at protein level by immunohistochemistry, studying several characteristic proteins like Ki67, E2F2 and pRB. This study was completed with the characterization of those protein in retinoblastoma cell lines.

This work was well performed and is very interesting, I have some commentaries for better understanding of the results:

1) Explanation of the meaning of gene acronyms, such as those mentioned in the text (Lines 435-440 and Fig. 5B): HDAC, FN1, BAAT, HIF1A, PRKAA2 (AMPKalfa), GUCA1A and their function in metabolic pathways.

2) From which patients the normal retina was obtained?

3) In Figure 5A are marked the predominant metabolisms: fatty acids, amino acids and pentose phosphate. Compared with Figure 4D heatmap all the results (Fig 4D, Fig. 5A/B) agree for fatty acids as is stated in the text (line 426/427). However, not all amino acids predominate in humor (Fig 4E and line 426/427).

4) Line 434 and Figure 5B: Valproic acid is a drug used in brain diseases and it is derived from naturally occurring valeric acid. How can the authors explain its presence in human’s tissues?

5)Taurine is downregulated (line 436 and Fig. 5B), however taurine metabolism is enriched (Fig. 4E). For example: BAAT gene (bile and CoA amino acid [glycine, taurine] N-acyl transferase) is highly expressed (line436 and Fig. 5B) but taurine is in low abundance (line 437 and Fig. 5B).  How can be explained these results?

6) High abundant metabolites like salicylic and myristic acids (line438) interact with downregulated genes (line 439 and Fig. 5B). These results may be explained as the non-use and accumulation of these metabolites due to the low expression of the involved genes?

7) Figure S2B: “Correlation analysis followed by clustering showed the relationship between the Rb subtypes: advanced Rb group correlate positively with each other (red). Non-advanced Rb showed no correlation or negative correlation” This is not what is seen in the figure: Advanced and non-advanced groups showed similar results, correlation with each other but not with the normal retina where they showed no correlation or negative correlation.

8) “Validation of molecular signature that specify clinical and histopathological grades of Rb”. Line 477/478 and Figure 6F, H: “RB1 and RBL2 expression was significantly low in bilateral Rb and groups D and E compared to pediatric controls”. The results of these figures show that RB1 expression was significantly low in unilateral Rb but not in bilateral and in groups D and E, in these tumors the expression is not significantly low (Fig. 6F). Expression of RBL2 in bilateral, unilateral Rb and group D and E was significantly lower than in controls but the results show a great dispersion and it is difficult to see any difference with normal retina (Fig. 6H).

9) The statement in lines 533-537 about alteration in sphingolipid and branch-chain amino acids metabolisms, apoptosis and DNA repair between RB1 null cells and RB1 complemented cells, with reference to Figure S3D is not clear. In the legend of the figure is not stated how can be differentiated both types of cells.

10) Discussion, Lines 575, 680 and 588 the word amplification is not correct, expression should be used instead since the analyses determined RNA transcripts not the number of  DNA copies.

11) Lines 585-586: The statement:”elevation of tumor-specific transcription factors like RUNX1, GABPB1, MYCN…their distinct transcriptional regulation in Rb tumors is a new observation”  is not completely correct because MYCN amplification and expression in Rb tumors has been reported in many papers such as those of Gallie group in Toronto.

Reviewer 2 Report

Congratulation for you.

The achievement from your team deserves recognition.

Many readers shall learn so much from your manuscript.

Thanks a lot.          

Reviewer 3 Report

The manuscript describes a summary of the authors’ extensive studies regarding metabolic characteristics of human retinoblastoma and vitreous tissues using metabolomics, gene expression, pathway/network, and immunohistochemistry analyses. The result that glycolysis seemed to be downregulated in RB tissue is interesting. Overall, the obtained data from these studies may be potentially useful for further scientific and medical researches as providing basic metabolic characteristics of RB.

Some minor points are raised to be addressed.

  1. 310: “Figure E” should be “Figure 2E”.
  2. 352: “Figure 3C” is not cited in the text.
  3. 397: “high” should be “higher”.
